# Real-Time Recognition and Detection of *Bactrocera minax* (Diptera: Trypetidae) Grooming Behavior Using Body Region Localization and Improved C3D Network

**DOI:** 10.3390/s23146442

**Published:** 2023-07-16

**Authors:** Yong Sun, Wei Zhan, Tianyu Dong, Yuheng Guo, Hu Liu, Lianyou Gui, Zhiliang Zhang

**Affiliations:** 1School of Computer Science, Yangtze University, Jingzhou 434023, China; 2Jingzhou Yingtuo Technology Co., Ltd., Jingzhou 434023, China; 3College of Agriculture, Yangtze University, Jingzhou 434023, China

**Keywords:** behavior recognition, 3D CNN, grooming behavior, *Bactrocera minax*

## Abstract

Pest management has long been a critical aspect of crop protection. Insect behavior is of great research value as an important indicator for assessing insect characteristics. Currently, insect behavior research is increasingly based on the quantification of behavior. Traditional manual observation and analysis methods can no longer meet the requirements of data volume and observation time. In this paper, we propose a method based on region localization combined with an improved 3D convolutional neural network for six grooming behaviors of *Bactrocera minax*: head grooming, foreleg grooming, fore-mid leg grooming, mid-hind leg grooming, hind leg grooming, and wing grooming. The overall recognition accuracy reached 93.46%. We compared the results obtained from the detection model with manual observations; the average difference was about 12%. This shows that the model reached a level close to manual observation. Additionally, recognition time using this method is only one-third of that required for manual observation, making it suitable for real-time detection needs. Experimental data demonstrate that this method effectively eliminates the interference caused by the walking behavior of *Bactrocera minax*, enabling efficient and automated detection of grooming behavior. Consequently, it offers a convenient means of studying pest characteristics in the field of crop protection.

## 1. Introduction

Pest control research has long been one of the top priorities in agriculture. Grooming behavior has important physiological significance and social functions in insects [1]. The study of insect grooming behavior offers a deeper understanding of their habits and carries important theories for pest control, enhanced crop yields, and protection of vegetation.

Grooming is an important behavior used by many animals to maintain healthy survival. It represents a large proportion of the animal’s total workout, accounting for 30–50% of its waking time [2]. Grooming behavior has the vital function of removing foreign bodies from the body surface [3]. Insect grooming is a robust innate behavior involving multiple independent movement coordination [4]. Despite the diversity in insect species, the main functions of grooming behavior are strikingly similar [5]. These functions primarily aim to reduce pathogen exposure [6], clean the cuticles, remove body secretions and epidermal lipids [7,8], collect fallen pollen as food [9], and eliminate potential immune threats from their body surfaces [10]. It can also be used to prevent dust or debris from inhibiting essential physiological functions (e.g., vision, reproductive behavior, or flight) by covering important body structures in a way that interferes with essential physiological functions (e.g., vision, reproductive behavior, or flight) [11]. It has also been suggested that the grooming behavior of insects may be an important mechanism of resistance to hostile organisms [12]. Therefore, studying grooming behavior holds significant value in understanding insect habits, providing insights into motor sequences of organisms [13], and establishing the pharmacological basis of insect grooming behavior [5].

As an oligophagous pest, *Bactrocera minax* lays eggs on almost all citrus crops and young fruits [14,15,16,17]. The fruits gradually rot and deteriorate, making them inedible or unsaleable. Data show that the economic loss caused by the *Bactrocera minax* is up to 300 million yuan annually. The rotting citrus also harms human health [18], thus this pest has become a significant concern in citrus-growing areas of China [19]. Studying the behavioral characteristics of grooming is an effective way of controlling *Bactrocera minax* [20]. It also aids in the development of safe and efficient control strategies [21].

In recent years, significant progress has been made in artificial intelligence techniques. They are highly sought after in various fields for their convenience, efficiency, and time, labor, and cost savings compared with traditional techniques [22]. Computer vision technology [23] combined with artificial intelligence has been widely used to reduce manual labor intensity or to even partially replace manual labor [24,25,26,27,28,29]. Compared with manual recognition, target detection algorithms [30] like YOLO [31], Faster R-CNN [32], SSD [33], and Transformer [34] have been able to achieve double improvement in terms of recognition accuracy and recognition efficiency. In insect behavior research, manual frame-by-frame determination remains the prevalent method for statistical analysis. However, factors like rapid behavior switching and personnel fatigue hinder desired results. To address these challenges, quantifying insect behavior using computer vision techniques is gaining prominence. By applying these tools, entomologists can uncover new insights and find answers to crucial questions [35].

Ji [36] first proposed using 3D convolutional neural networks for human behavior recognition. 3D convolutional neural networks have the advantage of simple network structure and can be executed end-to-end [37]. The subsequent C3D (Convolutional 3D Network) [38] model that emerged has also achieved better results in human behavior recognition. Hence, this study intends to extend C3D to creatures that move more frequently than humans and have smaller bodies, for example, *Bactrocera minax*. We propose a method based on the combined use of target region localization and a 3D convolutional neural network to recognize the grooming behavior of *Bactrocera minax* while ensuring that it can move freely. The YOLOv3 algorithm is used to obtain the bounding box of the body region of *Bactrocera minax* from video input of continuous frames. The distances between the center points of the bounding boxes of adjacent frames are then compared with the set threshold. The sizes and locations of the ROIs intercepted during grooming behavior (head grooming, foreleg grooming, fore-mid leg grooming, hind leg grooming, hind mid leg grooming, and wing grooming) are the same. The ROI frames are stacked in succession and fed into the 3D convolutional neural network model, enabling the recognition of grooming behavior in *Bactrocera minax* videos. This approach facilitates automatic detection and statistical analysis of insect grooming behavior over a specific period, introducing a novel and more efficient method of insect behavior research.

## 2. Materials and Methods

### 2.1. Experimental Materials

#### 2.1.1. Experimental Equipment and Environment

In this study, we identified and counted the grooming behavior of *Bactrocera minax* as a research objective. The hardware used was Intel core I9 9900K, 32 GB RAM, RTX 2080Ti GPU. We used Python 3.6, NumPy 1.19.2, OpenCV 3.4.2, Pillow 8.2.0, and Matplotlib 3.3.4 as the software development environment. The insect body region localization module was implemented using PyTorch, while the 3D convolutional neural network was constructed using the Keras library.

#### 2.1.2. Experimental Data Acquisition

A video of *Bactrocera minax*’s behavior was recorded using a Sony FDR-AXP60 video camera at a resolution of 1920 × 1080, a frame rate of 25 fps, and a filming duration of 60 min per adult. *Bactrocera minax* was placed in a petri dish (size 35 × 20 mm) in which it could move freely. Figure 1a shows the recording method and Figure 1b shows the raw video recording.

### 2.2. Experimental Methods

#### 2.2.1. General Flow Chart of the Methodology

To better illustrate the workflow of our approach in this article, we have created the following flowchart in Figure 2.

#### 2.2.2. Target Region Localization

A high-definition camera recorded the dataset with 1920 × 1080 shootinsg resolution and a wide camera field of view. Most scenes, other than the body region, were not meaningful for the experiment, and light changes or shadows in the scenes could also introduce errors in the experimental results. Thus, truncating the effective activity region of *Bactrocera minax* reduced the computation of the subsequent 3D convolution algorithm and improved the accuracy of behavior detection.

The YOLOv3 algorithm is an end-to-end target detection model that can output target bounding box location, confidence, and category information after inputting the image data to be measured. The structure of the YOLOv3 network, which uses Darknet-53 as the backbone extraction network, is shown in Figure 3. After the convolutional layer, the model incorporates a BN layer and a LeakyReLU activation function layer. Additionally, the use of residual skip connections helps reduce the model’s parameter count and enhances network convergence speed while maintaining the quality of feature extraction.

Compared with YOLOv2, YOLOv3 introduces the concept of an FPN (Feature Pyramid Network) for multiscale fusion. It achieves this by merging features from three different scales through upsampling and connection layers. Moreover, during the sampling process, a convolutional layer with a stride of 2 is employed to mitigate the loss of information from small targets caused by pooling.

In this study, the YOLOv3 target detection algorithm was used to detect and locate the body region of *Bactrocera minax*. The ROI required for the experiment was determined based on the bounding box coordinates calculated using the YOLOv3 algorithm. The ROI localization method diagram is shown in Figure 4.

#### 2.2.3. Target ROI Acquisition

The bounding box’s four vertex coordinates ($x_{1}$,$x_{2}$,$\;x_{3}{,x}_{4}$) were obtained frame by frame through object region localization. The coordinates of the center point ($X_{center}$, $Y_{center}$) of the bounding box of each frame can be calculated using Equations (1) and (2).
(1)$$X_{center}=x_{1}\left[0\right]+\frac{\left(x_{2}\left[0\right]-x_{1}\left[0\right]\right)}{2}$$
(2)$$Y_{center}=x_{1}\left[1\right]+\frac{\left(x_{4}\left[1\right]-x_{1}\left[1\right]\right)}{2}$$

The relative location changes that occur in the front mid-hind leg when *Bactrocera minax* performs grooming behavior also result in inconsistent sizes and locations of the bounding boxes predicted by YOLOv3 in adjacent frames. In other words, if the bounding box obtained by YOLOv3 is used directly as the ROI, the location and size of the region of interest between successive frames will cause field of view jitter, which is not conducive to the subsequent use of 3D convolution detection and modification behavior. Therefore, in this experiment, a judgment condition is added such that the *Bactrocera minax* remains within a bounding box of the same size and location as the ROI between consecutive frames when it is not walking. This judgment condition involves calculating the Euclidean distance (Equation (3)) of the center point of the original bounding box (obtained by direct prediction of YOLOv3) before and after the frame. If the distance is less than the set threshold, the coordinate point of the bounding box of the latter frame is replaced by that of the former frame, which generates the ROI.
(3)$$d_{ij}=\sqrt{{\left(x_{i}-x_{j}\right)}^{2}+{\left(y_{i}-y_{j}\right)}^{2}}$$

In another case, when the Euclidean distance between frames is greater than the threshold, the *Bactrocera minax* may be walking, thus the above method cannot guarantee that an ROI of equal size will be obtained. The flow chart of ROI acquisition is shown in Figure 5.

#### 2.2.4. Training Set Generation

This study identified six grooming behaviors of *Bactrocera minax* by improving the behavioral recognition C3D network. The experimental dataset was obtained by recording 20 adult *Bactrocera minax* for a total of 1200 min, with 60 min of filming per *Bactrocera minax*. Based on the professional classification of *Bactrocera minax* behavior, the dataset was divided into eight behaviors: head grooming, foreleg grooming, fore-mid leg grooming, hind leg grooming, mid-hind leg grooming, wing grooming, resting, and walking. The first six are grooming behaviors (Figure 6) while the last two are daily behaviors of *Bactrocera minax* and must be identified and excluded when performing video analysis. The videos in the dataset were then manually observed for the eight behaviors; videos in each behavior category were cropped to 7–10 s in length so that the experiment had short videos of the eight *Bactrocera minax* behaviors. The number of short videos is shown in Table 1.

For all feature videos, vertex coordinates of the bounding box were obtained using the method described in Section 2.2.2. ROIs in the same short video were then resized to the same size and cropped using the method described in Section 2.2.3. (ROIs of the same size and location could not be obtained in the case of walking behavior, but we could always ensure that the body of *Bactrocera minax* was in the ROI).

In this experiment, the two non-grooming behaviors—resting and walking—were classified into one category, resulting in 7 categories of behavioral feature videos containing only the body region of *Bactrocera minax*. These 7 categories of ROI feature videos were used as the training set in the 3D neural network in subsequent experiments. Next, the classified feature video folders were named head, foreleg, fore-mid leg, hind leg, mid-hind leg, wing, and 0 (0 indicates the two non-grooming behaviors—resting and walking). The feature videos in the class folders were renamed using the behavior name + number for easy management and labeling. The video names were stored with corresponding behavior number labels in a .txt file format. The training set data were indexed by reading the .txt files during training.

#### 2.2.5. Improved C3D Network Training

Two-dimensional convolutional neural networks have matured in image classification and target detection fields. However, the output of neural networks based on two-dimensional convolution only contains spatial information, and there is no representation of temporal information. The behavior of *Bactrocera minax* over time involves both temporal and spatial aspects, thus an additional dimension of information is needed, i.e., three-dimensional convolution. The convolution and pooling operations of the 3D convolutional neural network are performed at the spatiotemporal level, which is good for the extraction of temporal and spatial features. The 3D convolutional operation formula [36] is as follows:(4)$$y_{mn}^{wht}=relu(b_{mn}+\sum_{l}\sum_{i=1}^{W_{m}}\sum_{j=1}^{H_{m}}\sum_{k=1}^{T_{m}}v_{mn}^{ijk}u_{(m-1)l}^{\left(w-i)(h-j)(t-k\right)})$$
where $y_{mn}^{wht}$ denotes the output of the $N_{th}$ feature block in the $M_{th}$ layer at location (w,h,t) after the neuron operation, $v_{mn}^{ijk}$ denotes the weight of the nth feature block in the $M_{th}$ layer at location (i,j,k), and *b_mn_* denotes the bias of the nth feature block in the $M_{th}$ layer. The pooling layer can effectively reduce the size of the parameter matrix, thus reducing the number of parameters in the final fully connected layer, speeding up the computation and preventing overfitting.

In short, the video consists of consecutive image frames and the input video cube is formed by stacking several consecutive image frames. Convolutional operations obtain the spatiotemporal features in the image sequence through several different stereo convolutional kernels sliding in fixed steps, which finally determine the behavior generated in the video.

C3D is used as a three-dimensional convolutional neural network structure for behavior recognition and consists of eight convolutional layers: five maximum pooling layers, two fully connected layers, and a SoftMax layer. All three-dimensional convolutional kernels are 3×3×3 and have a step size of 1×1×1. The three-dimensional pooling layers are 2×2×2 (except the first pooling kernel, which is 1×2×2) and the network input shape is 16×112×112. Each fully connected layer consists of 4096 output units. The C3D network structure is shown in Figure 7.

We modified C3D to be more suitable for recognizing *Bactrocera minax*‘s behavior. The number of network layers was expanded to better extract deep features and improve network learning ability. The backbone structure of the network in this study was increased from 8 to 12 convolutional layers. However, if we only deepen the network, important features may be lost or overfitted. The number of parameters was much higher for 3D convolutional neural networks than for 2D convolutional neural networks. A deeper network requires more power. Therefore, it is not cost-effective for experiments to deepen the network without significantly increasing accuracy. In this study, we proposed a multi-scale fusion 3D convolutional neural network to recognize the grooming behavior of *Bactrocera minax*. The network structure is shown in Figure 8. The high-level semantic information obtained from the deep neural network is included in behavior recognition, and the shallow feature information can improve the robustness of the detection model. Fusing shallow feature information can help the high-level semantic information classify those behaviors with low differentiation, making it more suitable for the *Bactrocera minax*, which has a small size and high similarity of partial grooming behaviors. There were only 16 convolutional kernels in this convolutional layer, much fewer than the channels in the other input feature maps, thus the number of channels in the output feature map was also smaller. Thus, the number of parameters in the final, fused, fully connected layer was greatly reduced. The pooling layer following this fused convolutional layer differed from other pooling layer types. The backbone network uses 3D maximum pooling to improve the robustness of the extracted features and reduce the influence of useless information. In contrast, the fused layer uses a 3D average pooling layer because the input data only performs a little convolutional operation. The feature map is rich in time dimension information, texture, and background feature information. This experiment must incorporate this information in the fully connected layer to improve recognition accuracy, thus, compared with maximum pooling, average pooling can minimize the omission of shallow information. Finally, the behavior categories are classified based on the SoftMax layer.

#### 2.2.6. Identifying the Behavior of *Bactrocera minax* in Consecutive Frames

The proposed method identifies the grooming behavior of *Bactrocera minax* at each time point based on the frame or time order and counts the duration of the behavior. Since each grooming behavior of *Bactrocera minax* lasts at least 0.6 s, 16 consecutive frames (0.6 s) were selected as the unit of detection of grooming behavior in this experiment to ensure that grooming behaviors that occur for short durations are not missed or detected incorrectly. A consecutive sequence of 16 frames was taken, regions of interest (ROIs) extracted, grooming behavior recognition conducted, and the start and end times recorded. This process was repeated until the end of the video. When consecutive detection units were identified as having the same grooming behavior, the two units were merged when counting the duration interval and start/stop times. The start time, end time, and behavior type of each grooming behavior (including the occurrence of movement behavior) of *Bactrocera minax* were recorded in real time in a table generated during the recognition and detection process.

## 3. Results

### 3.1. Input Data Pre-Processing

ROI feature videos with seven types of behaviors were obtained using the dataset production method described in Section 2.2.4. First, the ROI feature video dataset was converted, frame-by-frame, into frame images. The converted dataset was split into training and test sets in a 4:1 ratio. Additionally, the frame image size was standardized to 171×128. To improve model accuracy and enhance the stability of the model while inputting 3D convolutional neural network data, this experiment converted the 171×128 pixel frame input size. The image was randomly cropped to 112×112 pixels and a continuous frame image was generated. A 3D sliding window was used to select 16 consecutive frames from this pixel region, which were then input into the network.. Before the data is input into the network, enhancement of each input data was performed using a horizontal flip of the image, with a probability of 0.5, and mean subtraction was performed along the three RGB channels of the frame image, to get an input network data size of 112×112×16×3.

### 3.2. Three-Dimensional Neural Network Training Parameter Setting

In this experiment, the 3D neural network was trained through 20 iterations; the batch size used for each training was 8. The SGD optimizer was chosen because it can guarantee faster training speeds for large-scale datasets. The initial learning rate was set to 0.003 and decreased to 1/10 of the original rate every 4 training iterations. The random deactivation rate of the dropout layer following the fully connected layer was set to 0.5. This way, some neural network units could be temporarily discarded from the network based on a certain probability, weakening the correlation between neuronal nodes and enhancing the generalization ability.

### 3.3. Continuous Frame Target Region Interception

Extracting continuous frames of the insect’s body region (ROI) represents a crucial step in this experiment. The stability of the ROI across consecutive frames significantly influences the accuracy of grooming behavior predictions. Therefore, establishing an appropriate threshold value is necessary.

Too small a threshold can cause frequent jitter of captured ROIs in successive frames. Although the stability of inter-frame ROIs can be ensured during walking if a larger threshold is selected, insect movement prevents the body from being completely within the ROI, resulting in the inaccurate judgment of walking behavior, which further affects the detection accuracy of walking behavior in this experiment. Therefore, it is necessary to find a threshold that guarantees the stability of the interception position of the region of interest between successive frames during grooming behavior, while the body is completely within the region of interest.

Threshold selection experiments were conducted to select experimental data from 7 categories of uncropped behavioral videos (including mobile behaviors) by randomly selecting 20 videos from each category. We selected thresholds of 10–200 intervals for ROI acquisition. We combined the grooming behavior ROI stability rate and the torso coverage rate during mobile behavior to select thresholds more suitable for the experiment. The frequency of ROI changes between frames of grooming behavior was calculated to determine the stability rate (Equation (5). The ROI coverage rate during walking behavior was also calculated (Equation (6). The experimental results are shown in Table 2.
(5)$${ROI}_{stability}=\frac{{Count}_{all}-{Count}_{change}}{{Count}_{all}}$$
(6)$${MOVE}_{cover}=\frac{{Count}_{all}-{Count}_{out}}{{Count}_{all}}$$

The results in the Table show that a threshold value between 60 and 80 can indicate the ROI stability rate of grooming behavior and the torso coverage rate during walking behavior. In this experiment, 80 was chosen as the final threshold value because a larger threshold value can make the size and location of the intercepted ROI during walking more stable.

During the video recording process, we found that *Bactrocera minax* often exhibited walking behaviors that were accompanied by fore-mid, hind-leg, and body movements. Based on the method proposed in this study, the grooming behavior was detected more efficiently and accurately by cropping the ROI, considering that *Bactrocera minax* does not move during grooming. However, the inconsistency in the size and location of the cropped ROI between frames during walking could not be avoided. To better determine the walking behavior, two schemes were compared in this experiment. The first approach is to not perform ROI cropping on the feature dataset of walking behaviors during dataset creation. Instead, the original-sized walking behavior dataset is directly used for model training. Considering that the features of walking behavior are easier to learn than those of grooming behavior, this smoother original-size training set may have some effect on the inter-frame ROI inconsistency in the consistent video to be detected and has some generalization. The other way is using the cropped ROI dataset (with possible inconsistency in ROI size and location in consecutive frames) as the model training set for the walking behavior of *Bactrocera minax*. The accuracy of the model trained using these two schemes on the test set is shown in Table 3.

### 3.4. Statistical Results of Grooming Behavior Identification and Detection of Bactrocera minax

In this experiment, after preprocessing 1896 feature videos, 29,253 behavioral intervals were obtained. Each behavioral interval consists of 16 frames of images. The training set and test set were divided into 22,315 and 6938 behavior intervals, respectively. The network trained 20 epochs, with a total training time of about 40 h. The accuracy rates of the training set and test sets during the training process are shown in Figure 9a. The accuracy curves leveled off after nine iterations, giving the best results and model convergence. The accuracy rate in the test set stabilized at about 93.45%. Figure 9b shows the loss curve during model training. The loss curve of the test set had some fluctuations at 9–11 Epochs, probably due to some noise or extremely special poses in the data in the test set. The overall loss curve trended from decreasing to leveling off, indicating that the training was effective.

We selected another 700 pre-processed behavioral video models to evaluate the effect of behavioral classification. Each video lasted 8–10 s. The model was evaluated using test set accuracy, recall, and F1 scores [39]. Accuracy and recall rate were calculated as shown below.

Accuracy is the simplest and most intuitive evaluation metric for classification problems. Accuracy reflects the proportion of all prediction outcomes that the model correctly predicts. However, when the target sample is unbalanced, the largest proportion of the target sample can significantly impact accuracy.
(7)$$Accuracy=\frac{TP+TN}{TP+TN+FP+FN}$$

*Precision* is the proportion of samples for which the model predicts true positives to those predicted to be positive, reflecting the model’s accuracy in classifying a class.
(8)$$Precision=\frac{TP}{FP+TP}$$

*Recall* is the ratio of the number of samples identified as true positives to those that are positive and reflects the ability of the algorithm to find all positive samples.
(9)$$Recall=\frac{TP}{FN+TP}$$

The *F* Score is a combination of *Precision* and *Recall*. When we consider *Precision* and *Recall* as equally important, it is called the *F*1 Score. *F*1 is calculated as follows:(10)$$F1=\frac{2\times Recall\times Precision}{Recall+Precession}$$

The following results were obtained from the experiments. The model’s accuracy was 93.46%. The results of the behavior classification for each category are given in Table 4.

The true positive and false positive rates for each category of behavior recognition can be referred to the confusion matrix in Figure 10.

As can be seen, the model gave good results in all behavioral classifications, apart from mid-hind leg and wing grooming. After examining the original video of the predicted incorrect behavior, we speculate that *Bactrocera minax* changed their behavior within a 16-frame time window or multiple behaviors occurred, resulting in failure of behavior recognition.

To verify the usability of our method, we selected 15 unedited videos of *Bactrocera minax*. Professionals first counted the type and duration of *Bactrocera minax* grooming behavior before using the model. The results of the two methods were compared.

The start time and end time of each behavior were recorded through manual observation and statistical analysis. Different behavior types occurring within the same period but with duration intervals greater than 25 frames were considered as showing a difference. The comparison method is outlined in Figure 11, and the results of the difference-degree calculation are presented in Table 5. The average difference degree between this method and manual recognition is approximately 12%. However, the method offers a significant advantage in terms of speed, being 3–5 times faster than manual observations, thereby demonstrating practicality.

### 3.5. Comparison between Methods

Five grooming behavior detection methods were compared on the test set, including Zhang’s method [21], based on spatiotemporal context, and Zou’s method [20], which utilizes keypoints recognition with DeepLabCut. The third and fourth methods involved experimenting with different data processing approaches, while directly employing the C3D network [40]. The final results indicate that the method proposed in this study can achieve real-time detection and analysis of grooming behavior in *Bactrocera minax* videos while maintaining accuracy. The experimental results of different methods are presented in Table 6.

## 4. Discussion

The study of insect behavior holds both theoretical and practical significance in the utilization of beneficial insects and in pest control [41]. As insect behavior analysis continues to advance, statistical analysis of insect behavior categories over a specific timeframe has become integral to fundamental research. To enhance the efficiency and accuracy of insect behavior identification, this study proposes a mechanism that combines computer vision technology and deep learning; specifically, a three-dimensional convolutional neural network is employed for insect grooming behavior recognition. During the experiment, the research team collected various insects, including *Bactrocera minax*, *Bactrocera cucurbitae (Coquillett)*, *Procecidochares utilis*, and *Bactrocera dorsalis Hendel*, from Jingzhou in Hubei Province, Kunming in Yunnan Province, and Haikou in Hainan Province. We compared the behavioral characteristics of these insects, ultimately discovering that *Bactrocera minax* exhibits higher activity levels, more frequent grooming behaviors, and more easily recognizable behavior. Therefore, *Bactrocera minax* was used as the main object of the experiment.

In recent years, bioinformatics analysis, which combines biological experimental data with computer technology, has found widespread applications in entomology [42]. This includes the utilization of deep learning methods for automatic insect counting [43] and various pest detection techniques [44,45,46]. Unlike traditional image processing methods, these approaches leverage deep learning techniques to construct neural networks that offer enhanced prediction generalization and improved feature extraction in different environmental contexts. Automatic insect counting and pest detection focus more on spatial information and often use a two-dimensional convolution kernel to extract the required feature information. However, in behavior recognition, the time between successive frames is particularly important, thus this study chose to use three-dimensional convolution. Denis S. Willett [47] classified the feeding patterns of insects by automatically monitoring the voltage changes on the insect food source feeder circuit, an advance that greatly reduced the time and effort required to analyze the insects. This approach requires the placement of equipment on the insect’s body. Our team preferred to start from another perspective to identify and detect insect behavior in a non-invasive way, reducing the intervention of equipment on the insects and reducing operational costs.

There are limited methods available for detecting insect grooming behavior from videos. Zhang [21] proposed a method that combines spatiotemporal context and convolutional neural networks, achieving promising results. This approach generates spatiotemporal feature images by fusing temporal and spatial features and classifies behavior based on these images. Another method, presented by Nath [48], utilizes key point tracking (DeepLabCut) to estimate animal posture. DeepLabCut demonstrates robust performance, even with a small number of training sets and challenging backgrounds such as cluttered and unevenly illuminated conditions. Zou [20] applied body key point tracking to *Bactrocera minax* in their experiments. However, the key point tracking was disturbed by the fact that the body of *Bactrocera minax* is relatively small in the video and the body parts are obscured from each other during the behavior.

Convolutional neural networks (CNNs) have advanced behavior recognition, with 3D convolution expanding it to the spatiotemporal domain [40]. Inspired by the success of C3D in human behavior recognition, our team sought to apply it to insects. Initial attempts using the C3D structure and raw-size video data yielded unsatisfactory detection results. Scaling and random cropping of frame images, along with the limited appearance of *Bactrocera minax* in the dataset (7% of the field of view), resulted in significant feature loss and poor detection outcomes. To address this, our team proposed the idea of cropping insect body parts; subsequent experiments validated the feasibility of this approach.

After the final optimization of the network, the total recognition accuracy of the method used in this study to determine the grooming behavior of *Bactrocera minax* was 93.46% and the discrepancy with the manual recognition results was about 12%. When conducting the variance experiments, we found that most of the variances occurred when *Bactrocera minax* performed behaviors over very short durations that could be overlooked by human eyes. However, the method used in this study was able to recognize these behaviors. In the case of multiple rapid switching between foreleg grooming and head grooming, manual observation may record only one behavior, and such rapid switching between behaviors is common in the dataset in this experiment. 

The next direction that needs to be investigated is improving the accuracy rate while ensuring detection speed. We recommend the following three aspects: The first aspect is to expand the size of the training set by changing data with incorrect labels after manual comparison and adding these to the training set. The second aspect is the optimization of the structure of the network. For example, a deeper 3D CNN structure may help in the extraction of behavioral features. Hara [49] proposed that a deep network 3D ResNet applies to behavioral detection, where the 3D residual structure can effectively avoid the gradient explosion and gradient disappearance generated by the deep network. This structure can be added to the network structure of this experiment to extract deeper behavioral features. The third aspect can be initiated from the prediction result determination method. For example, the frame image input is divided into two branches. One branch of the model determines the results frame by frame using the method described in this article, while the other branch adopts a skipping frame method to determine the results. The skipping frame method better reflects the motion trend. Then, the final prediction result is obtained by combining these two results, allowing the model to fully learn the entire motion process.

## 5. Conclusions

This paper presents a statistical mechanism that combines insect body region localization and an improved C3D network for the recognition of *Bactrocera minax* grooming behavior. The proposed method involves precise cropping of *Bactrocera minax* body parts using the YOLO algorithm. The cropped consecutive frames are then stacked and inputted into the improved 3D convolutional neural network. Using scale fusion, the network combines deep and shallow features to perform classification and interval recording of grooming behavior. Experimental results demonstrate the method’s efficiency, accuracy, and real-time capability in detecting *Bactrocera minax* grooming behavior from videos, with errors falling within an acceptable range. This approach provides a novel means for analyzing insect behavior in the context of precision agriculture.

## Figures and Tables

**Figure 1 sensors-23-06442-f001:**
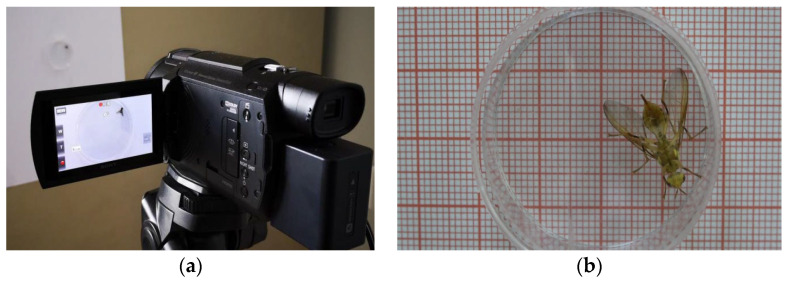
(**a**) The experimental data recording method; (**b**) original video effect.

**Figure 2 sensors-23-06442-f002:**
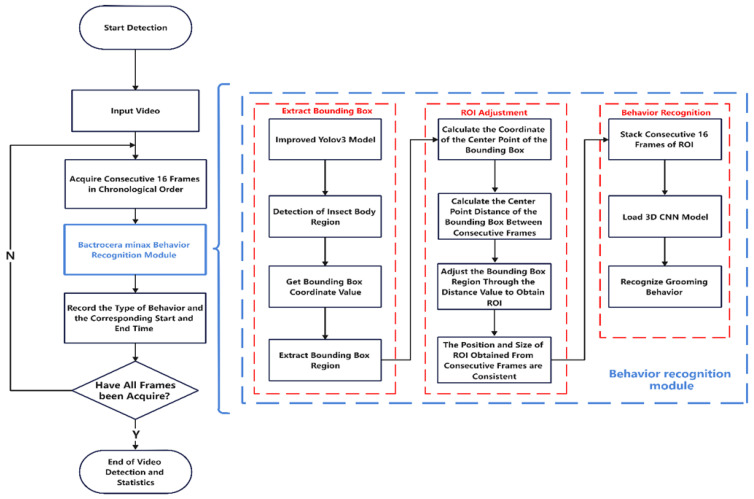
Flowchart of the overall methodology.

**Figure 3 sensors-23-06442-f003:**
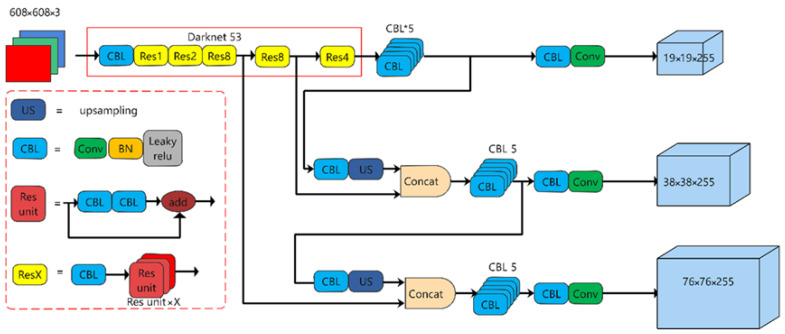
YOLOv3 network structure diagram.

**Figure 4 sensors-23-06442-f004:**
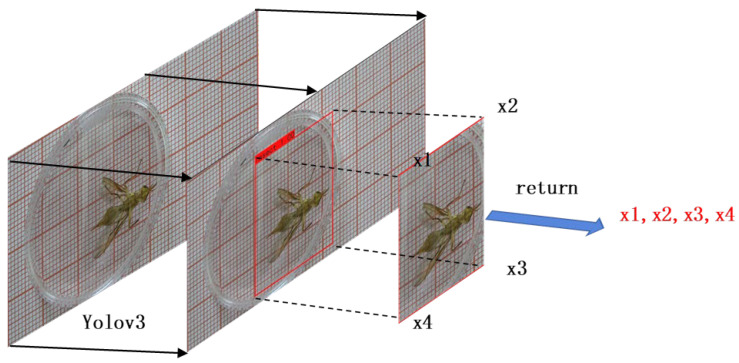
Schematic diagram of the regional localization method.

**Figure 5 sensors-23-06442-f005:**
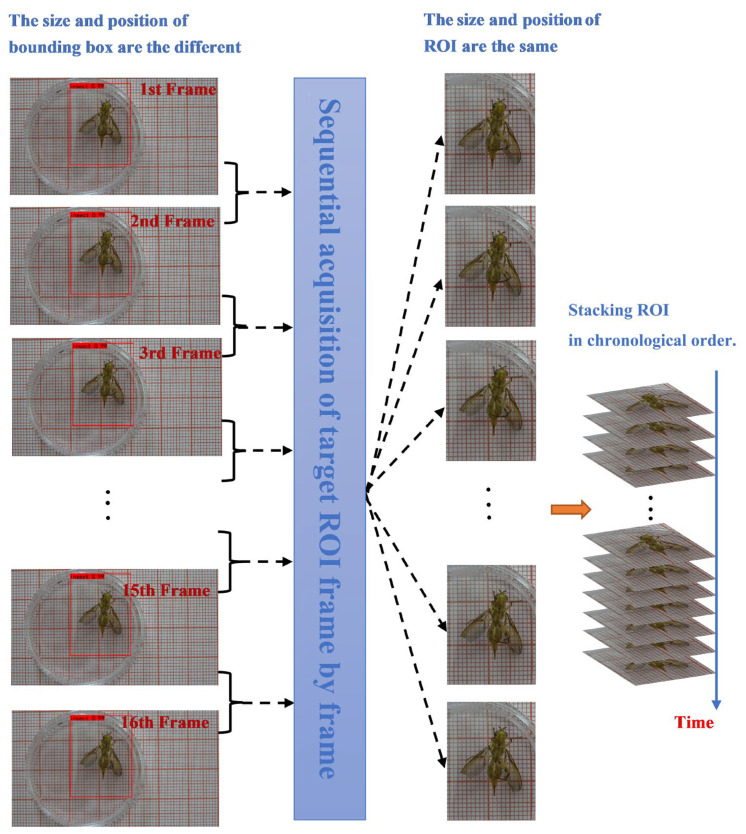
Grooming behavior ROI adjustment method.

**Figure 6 sensors-23-06442-f006:**
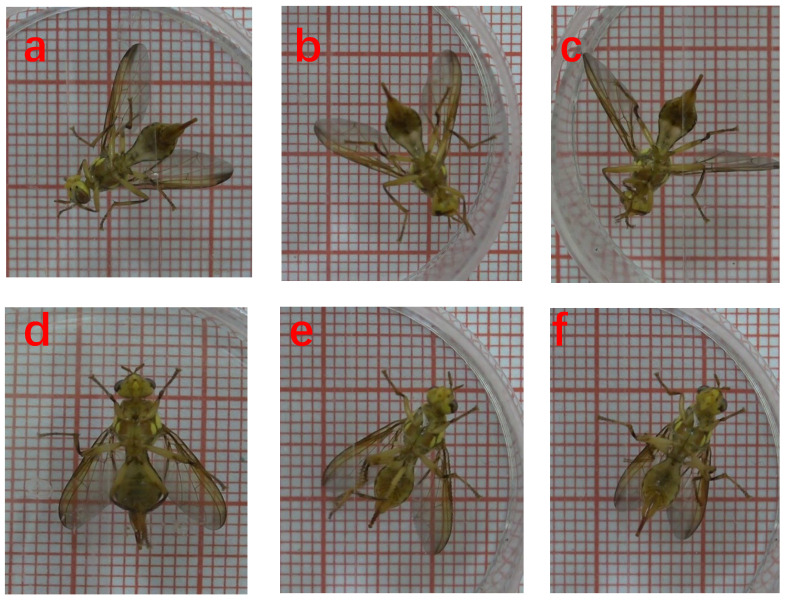
Six grooming behaviors of *Bactrocera minax*. (**a**) Head grooming; (**b**) Foreleg grooming; (**c**) Front-mid leg grooming; (**d**) Hind leg grooming; (**e**) Mid-hind leg grooming; (**f**) Wing grooming.

**Figure 7 sensors-23-06442-f007:**
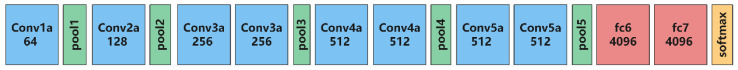
The CD3 network structure.

**Figure 8 sensors-23-06442-f008:**
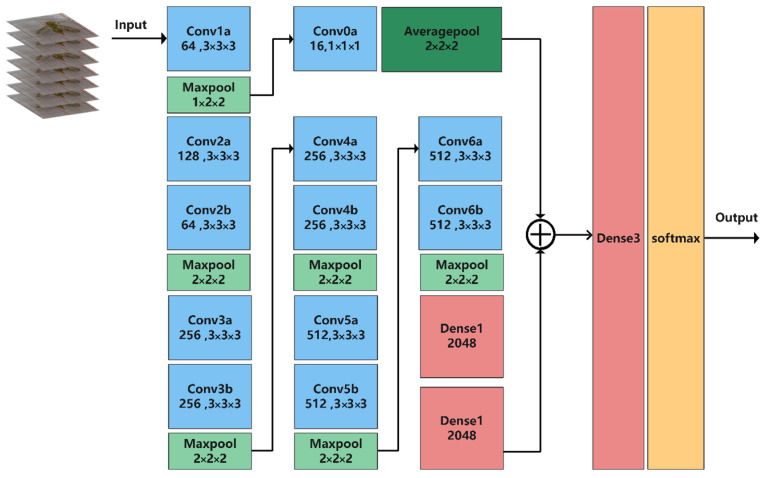
Network structure used in this study.

**Figure 9 sensors-23-06442-f009:**
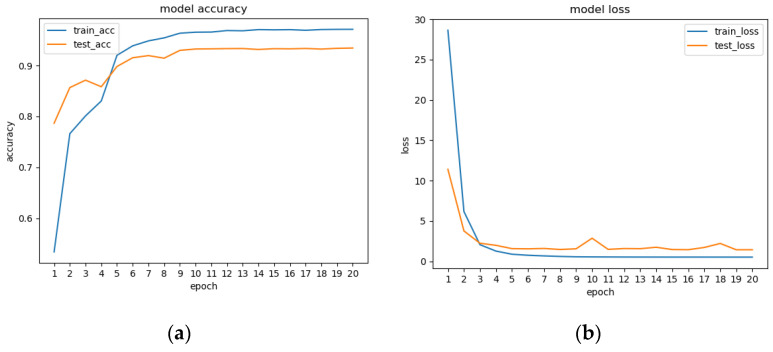
(**a**) Model training accuracy; (**b**) loss curves.

**Figure 10 sensors-23-06442-f010:**
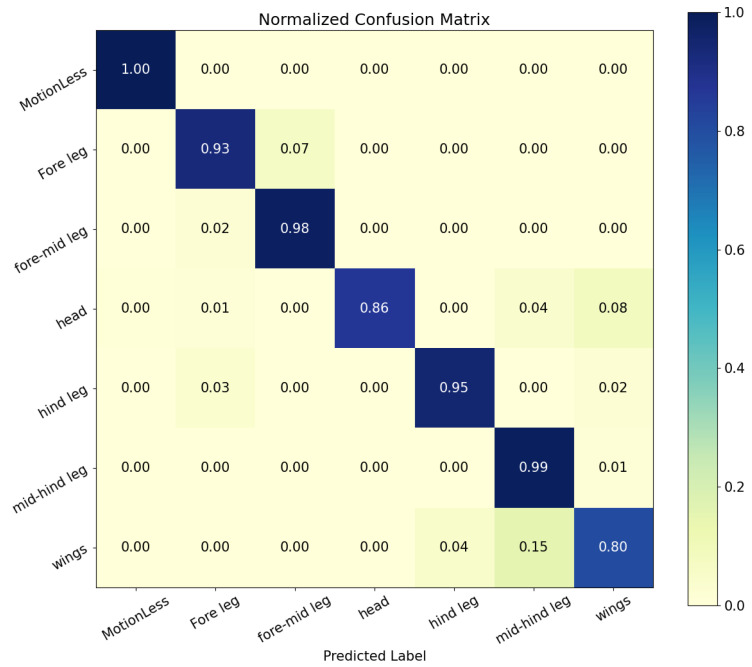
Confusion matrix of all predicted classes.

**Figure 11 sensors-23-06442-f011:**
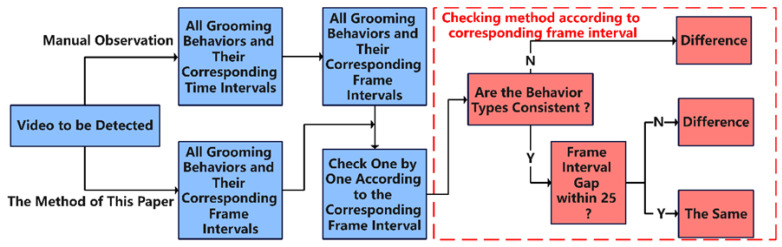
Comparison flowchart of the method used in this study versus manual observation.

**Table 1 sensors-23-06442-t001:** Number of feature videos of each behavior.

Type of Behavior	Quantity	Type of Behavior	Quantity
Head grooming	269	Hind leg grooming	272
Foreleg grooming	266	Wing grooming	266
Fore-mid leg grooming	298	Resting and walking	266
Mid-hind leg grooming	259		

**Table 2 sensors-23-06442-t002:** Experimental results of threshold selection during ROI acquisition.

Threshold	Grooming Behavior ROI Stability Rate	Body Coverage Rate of Walking Behavior	Threshold	Grooming Behavior ROI Stability Rate	Body Coverage Rate of Walking Behavior
10	95.8%	100%	110	100%	95.8%
20	98.0%	100%	120	99.9%	95.2%
30	98.6%	100%	130	100%	92.8%
40	97.9%	100%	140	100%	87.4%
50	99.5%	100%	150	100%	87.7%
60	99.9%	100%	160	100%	83.2%
70	99.9%	100%	170	100%	83.8%
80	100%	99.4%	180	100%	83.2%
90	100%	98.6%	190	100%	73.3%
100	100%	98.8%	200	100%	61.9%

**Table 3 sensors-23-06442-t003:** Accuracy of different training sets on the test set (walking behavior).

Walking Behavior Training Set Type	Accuracy
Make ROI Cuts	95.8%
No ROI cropping	89.7%

**Table 4 sensors-23-06442-t004:** Results of behavior classification for each category.

Behavior Category	*Precision*	*Recall*	*F*1 Score
Stationary	0.99393572	1	0.99695864
Foreleg grooming	0.9390681	0.92907801	0.93404635
Fore-mid leg grooming	0.93997965	0.97777778	0.95850622
Head grooming	0.99920319	0.86363636	0.92648689
Hind leg grooming	0.95076401	0.94594595	0.94834886
Mid-hind leg grooming	0.82645503	0.98611111	0.89925158
Wing grooming	0.79351032	0.80298507	0.79821958

**Table 5 sensors-23-06442-t005:** Differences between our method and manual observation.

Video Number	Number of Behaviors Recorded Using Manual Observation.	Number of Behaviors Accurately Recorded Using Our Method.	Degree of Difference
1	76	67	11.84%
2	36	31	13.88%
3	47	43	8.51%
4	67	60	10.44%
5	81	73	9.87%
6	39	35	10.25%
7	99	88	11.11%
8	104	88	15.38%
9	62	55	11.29%
10	67	60	10.44%
11	116	95	18.10%
12	48	41	14.58%
13	82	72	12.19%
14	56	48	14.28%
15	95	86	9.47%
Average	71.6	62.8	12.11%

**Table 6 sensors-23-06442-t006:** Comparisons between methods (input video frame rate measured at 25 fps).

Experimental Method	Grooming Behavior Accuracy	Detection Speed (Frames/s)
Temporal Context	93.43%	10.3
Deeplabcut	92.55%	7.8
C3D (without cropping)	76.82%	21.8
C3D (using ROI cropping)	86.11%	21.4
Our method	93.46%	20.3

## Data Availability

The data presented in this study are available on request from the corresponding author.

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
