# Peer review of "Real-Time Recognition and Detection of Bactrocera minax (Diptera: Trypetidae) Grooming Behavior Using Body Region Localization and Improved C3D Network"

_sensors, 2023, doi:10.3390/s23146442_

Round 1

Reviewer 1 Report (Previous Reviewer 2)

The article is devoted to solving the problem of video recognition. The topic of the article is relevant. The structure of the article is not classical for MDPI (Introduction, Models and Methods, Experiments, Discussion, Conclusions). The level of English is acceptable. The article is easy to read. The quality of the figures is acceptable. The article cites 48 sources, many of which are not relevant. The References section is sloppy.

The following comments and recommendations can be formulated on the material of the article:

1. First of all, the content of the article does not match its title. There is simply no "Statistical Mechanism" in the article. Moreover, all ten formulas given in the article are well known (references should be added). As far as I understand, "Real-time" is provided by high-speed shooting. The model used for recognition and detection is in no way adapted for "Real-time".

2. There is simply no scientific novelty in the work. The work offers nothing new in either Statistical or Real-time or Recognition and Detection. Moreover, I am sure that the system will not distinguish "Bactrocera minax" from an ant with wings. Thus, it is not an "Article". I think it's a Report.

3. Many questions arise about the rationale for the tools chosen by the authors. For example, if the authors say "Real-time", then I will answer "YOLO". The YOLO model directly predicts bounding boxes and class probabilities with one network in one estimate. The simplicity of the YOLO model allows for real-time predictions. Initially, the model takes an image as input. He divides it into an SxS grid. Each cell in this grid predicts B bounding boxes with a confidence score. This confidence is simply the probability of finding the object times the IoU between the predicted and ground truth fields. The CNN used is inspired by the GoogLeNet model, which introduces initial modules. The network has 24 convolutional layers followed by 2 fully connected layers. Reduction layers with 1x1⁴ filters followed by 3x3 convolutional layers replace the original seed modules. The Fast YOLO model is a lighter version with only 9 convolutional layers and fewer filters. Most of the convolutional layers are pre-trained using the Classified ImageNet dataset. Four convolutional layers are added to the previous network, followed by two fully connected layers, and it is fully retrained with the 2007 and 2012 PASCAL VOC datasets. The last layer outputs a tensor S*S*(C+B*5) corresponding to the predictions for each grid cell. C is the number of estimated probabilities for each class. B is a fixed number of anchor boxes per cell, each of these anchor boxes is associated with 4 coordinates (box center coordinates, width and height) and a confidence value.

4. The authors say “Ji [36] first proposed using 3D convolutional neural networks for human behavior 68 recognition. 3D convolutional neural networks have the advantage of simple network 69 structure, it can be executed end-to-end [37]. The subsequently emerged C3D [38] model 70 has also achieved better results in human behavior recognition.” I agree in principle with what was written, but why did the authors decide that a model that worked well in classifying human behavior would be effective in classifying insect behavior? Dear authors, the power of such neural networks lies in pre-training on colossal specialized datasets. In these datasets, insects were absent as a class.

-

Author Response

Dear Reviewer #1,

Thank you for your valuable feedback. We have made improvements to the unclear descriptions in the title and main text, hoping to reduce the inconvenience caused to readers. Please find our responses below:

Comment1:

First of all, the content of the article does not match its title. There is simply no "Statistical Mechanism" in the article. Moreover, all ten formulas given in the article are well known (references should be added). As far as I understand, "Real-time" is provided by high-speed shooting. The model used for recognition and detection is in no way adapted for "Real-time".

Response1:

We have revised the description of "statistical mechanisms" in the title. In our paper, we discussed the continuous frame count for different insect behaviors. We mistakenly referred to it as a "statistical mechanism," which is actually unrelated to statistics. We apologize for any confusion caused by this error. We have now removed the term "statistical mechanism."

In the revised version, we have included references to the 3D convolution and evaluation metrics sections. The formula for 3D convolution is based on the following literature:

[1] Ji, S.; Xu, W.; Yang, M.; Yu, K. 3D convolutional neural networks for human action recognition. IEEE transactions on pattern analysis and machine intelligence 2012, 35, 221-231.

Regarding evaluation indicators, the following literature was cited:

[2] Zhou, Z.-H. Machine learning; Springer Nature: 2021.

The other formulas in the paper are designed for the experimental scenario in this paper.

As for "real-time," our work aims to replace the manual recognition of grooming behavior. Compared to the method of manually observing videos frame by frame, our proposed method can fully automate the recognition and statistical analysis of grooming behavior. For instance, in a one-hour video (25 frames, 1080P), manual observation alone would take 5-8 hours, and the results would need to be cross-checked by three individuals. In contrast, our method, as described in this article, requires only about 1.2 hours and achieves the same level of accuracy for insect behavior research.

Our method has been successfully applied to real-time video capture, achieving a frame rate of over 20fps, which is close to the standard movie frame rate of 24fps. This enables a "real-time" intuitive experience for human viewers.

Comment2:

There is simply no scientific novelty in the work. The work offers nothing new in either Statistical or Real-time or Recognition and Detection. Moreover, I am sure that the system will not distinguish "Bactrocera minax" from an ant with wings. Thus, it is not an "Article". I think it's a Report.

Response1:

Our work is the design and experimental demonstration of the new detection method. The main content of our work is to automatically detect the grooming behavior and record the number of consecutive frames in which the behavior occurs (for example, from frame M to frame N, the insect has carried out forefoot grooming). Therefore, the behavior detection method proposed in this paper is applied to the grooming behavior of Bactrocera minax (defined by the specialty of insect behavior, and we have cited relevant references), rather than detecting the species of insects.

Neither the YOLO algorithm nor the C3D algorithm alone can fulfill the requirements of real-time detection of insect behavior. The YOLO algorithm is primarily used for target detection and lacks the capability of behavior recognition. While we utilized C3D to train the behavior recognition model, it was not effective in directly detecting grooming behavior in Bactrocera minax. Therefore, we implemented two improvement strategies to optimize the detection performance of C3D.

Firstly, we employed YOLO to extract the region of interest (ROI) for the insects and fed these clipped ROIs into the C3D network. This approach yielded significant improvements. Additionally, we enhanced the network structure of C3D and retrained the model, leading to further accuracy improvements. As a result, we achieved a behavior recognition accuracy of 93.68% and an average frame rate of 20fps.

Comment3:

Many questions arise about the rationale for the tools chosen by the authors. For example, if the authors say "Real-time", then I will answer "YOLO". The YOLO model directly predicts bounding boxes and class probabilities with one network in one estimate. The simplicity of the YOLO model allows for real-time predictions. Initially, the model takes an image as input. He divides it into an SxS grid. Each cell in this grid predicts B bounding boxes with a confidence score. This confidence is simply the probability of finding the object times the IoU between the predicted and ground truth fields. The CNN used is inspired by the GoogLeNet model, which introduces initial modules. The network has 24 convolutional layers followed by 2 fully connected layers. Reduction layers with 1x1⁴ filters followed by 3x3 convolutional layers replace the original seed modules. The Fast YOLO model is a lighter version with only 9 convolutional layers and fewer filters. Most of the convolutional layers are pre-trained using the Classified ImageNet dataset. Four convolutional layers are added to the previous network, followed by two fully connected layers, and it is fully retrained with the 2007 and 2012 PASCAL VOC datasets. The last layer outputs a tensor S*S*(C+B*5) corresponding to the predictions for each grid cell. C is the number of estimated probabilities for each class. B is a fixed number of anchor boxes per cell, each of these anchor boxes is associated with 4 coordinates (box center coordinates, width and height) and a confidence value.

Response3:

Thank you for your suggestion! We used YOLO to extract the ROI of Bactrocera minax movement, but this is only the first step of the method. YOLO, as a target detection algorithm, cannot be used for behavior recognition. In this case, we use YOLO to detect the single class target of the Bactrocera minax, and then take the detected boundary box as the ROI area, and take 16 consecutive images within the ROI area, input them into the improved C3D network for behavior detection.

Because this is not a simple target detection problem, our "real-time" is not completely dependent on YOLO. In this scenario, we choose YOLO because it is easy to use and has achieved good results. YOLO shows good performance when used for ROI extraction, but it can also be replaced by other target detection methods.

Comment4:

The authors say “Ji [36] first proposed using 3D convolutional neural networks for human behavior 68 recognition. 3D convolutional neural networks have the advantage of simple network 69 structure, it can be executed end-to-end [37]. The subsequently emerged C3D [38] model 70 has also achieved better results in human behavior recognition.” I agree in principle with what was written, but why did the authors decide that a model that worked well in classifying human behavior would be effective in classifying insect behavior? Dear authors, the power of such neural networks lies in pre-training on colossal specialized datasets. In these datasets, insects were absent as a class.

Response4:

The neural network model for human behavior recognition can not be directly applied to insects. Therefore, in order to recognize the behavior of Bactrocera minax, we took a large number of video data to train the neural network.

For human behavior recognition, a typical dataset is ucf-101, which is also the dataset used in the paper of human behavior recognition using C3D network. The data set is as follows:

  • Dataset Name: ucf-101 (2012)
  • Total Videos: 13320 videos
  • Total duration: 27 hours
  • Video source: YouTube capture
  • Video category: 101
  • It mainly includes five types of movements: human object interaction, simple limb movements, human interaction, playing musical instruments, and sports
  • Each category (folder) is divided into 25 groups, with 4-7 short videos in each group, and the duration of each video varies

For the grooming behavior of citrus fruit fly, we recorded a total of 1200 minutes behavior videos as a dataset. This is sufficient for grooming behavior recognition.

In addition, as you mentioned, C3D is not good enough when it is directly used for insect behavior detection. Therefore, we used YOLO to extract the ROI of the moving area and improved the network structure of C3D. In our experiment, it has also proved that our improvement is effective and can effectively recognize grooming behavior.

Once again, we sincerely appreciate your thorough review and guidance. We believe these revisions have significantly improved the quality and accuracy of the manuscript. Should you have any further concerns or suggestions, please do not hesitate to let us know. We look forward to your final evaluation of the revised paper.

Best regards,

Wei Zhan

Reviewer 2 Report (Previous Reviewer 3)

In writing the abstract the authors should be aware that the reader has not yet read the paper so the terminology needs to be better explained. 

“The overall recognition accuracy of each behavior reached 93.46%. We compared the statistical results obtained from the detection model with the manual observations, and the average difference was about 12%. Meanwhile, the detection efficiency of the method is 3-5 times higher than that of manual observation, which can meet the demand for real-time detection.”

Mentions “each behaviour”. Need to name the behaviours or mention that the intent is to rtecognise different grooming behaiors that involve the legs stroking each other or the head, wings or body.

What are “statistical results” and what does a 12% difference mean? This has to be clarified for the abstract.

What is detection efficiency? How can it be 3 to 5 times higher than manual detection?  If manual detection is the benchmark then it seems extraordinary that the algorithm can be 3–5 times more efficient. Does this mean that the algorithm detects 3 –5 more times grooming events than manual detection? Explain.

L 46 neeeds a comma, not a full-stop.

Throughout, ensure that the species name  is in italics

L 70. The full name of c3D should be stated at the first use of the acronym

L. 256 “combing” behaviour. Shold this be grooming behaviour?

L 375. Is reference to Table 4 and Figure 10 correct? I think it should be Table 5 and Figure 11

Table 5. The title needs to better describe the data in the table. I think the title could be “difference between automated and manual detection of grooming behaviors“. What does “number of matching results” mean. Remove ”manual statistics” from the title. Use “manual detections” instead. This section needs to be described better. 

This is a major improvement on the first submission. The English expression could be improved.

Author Response

Dear Reviewer #2,

Thank you very much for reviewing our manuscript and providing valuable feedback. We sincerely appreciate the time and effort you have dedicated to offering constructive comments.

Based on your suggestions, we have carefully reviewed the manuscript and made the necessary revisions. We strive to ensure the accuracy and completeness of our research, and we have corrected the errors you pointed out.

Please find our responses and the corresponding changes below:

In writing the abstract the authors should be aware that the reader has not yet read the paper so the terminology needs to be better explained.

Comment1:

In writing the abstract the authors should be aware that the reader has not yet read the paper so the terminology needs to be better explained.

Response1:

Thank you for your suggestion! In the abstract, we have now provided a clearer explanation of the terminology used.

Comment2:

Mentions “each behavior”. Need to name the behaviors or mention that the intent is to recognize different grooming behaviors that involve the legs stroking each other or the head, wings or body.

Response2:

Thank you for your suggestion! Regarding the mention of "each behavior", we revised the sentence to show the name of each grooming behavior.

Comment3:

What are “statistical results” and what does a 12% difference mean? This has to be clarified for the abstract.

Response3:

Thank you for your suggestion! "Statistical results" is the error description of the test results. It actually represents the detection result of the grooming behavior of the algorithm proposed in this paper.

Behavior occurs continuously over a period of time. For example, in a video, the Bactrocera minax has a head grooming behavior from frame 24 to frame 96. We recorded 15 videos of daily behavior of Bactrocera minax, and manually recorded the frame sequence of carding behavior of Bactrocera minax in the video. Comparing this result with the recorded results obtained by this method, the average difference is 12%.

When manually observing grooming behavior, three experts independently recorded their observations. The discrepancies among the three individuals ranged from 10% to 20%.

Comment4:

What is detection efficiency? How can it be 3 to 5 times higher than manual detection?  If manual detection is the benchmark then it seems extraordinary that the algorithm can be 3–5 times more efficient. Does this mean that the algorithm detects 3 –5 more times grooming events than manual detection? Explain.

Response4:

Thank you for your suggestion! The "detection efficiency" is 3-5 times that of manual detection, which represents the detection of the same 60-minute video. The running time of the algorithm in this paper is 20% to 25% of that of manual detection. We have corrected the description in the abstract.

Comment5:

L46 neeeds a comma, not a full-stop.

Response5:

Thank you for your suggestion! We have corrected the punctuation at L46, replacing the full-stop with a comma as suggested.

Comment6:

Throughout, ensure that the species name is in italics

Response6:

Thank you for your suggestion! Throughout the paper, we have ensured that the species name is consistently formatted in italics.

Comment7:

L 70. The full name of c3D should be stated at the first use of the acronym.

Response7:

Thank you for your suggestion! At L 70, we now state the full name of c3D (e.g., " Convolutional 3D Networks") at its first use and use the acronym thereafter.

Comment8:

  1. 256 “combing” behaviour. Should this be grooming behaviour?

Response8:

Thank you for your suggestion! We carefully checked the paper and replaced all the combining behavior in the paper with grooming behavior.

Comment9:

L 375. Is reference to Table 4 and Figure 10 correct? I think it should be Table 5 and Figure 11

Response9:

Thank you for your suggestion! We have verified the references, and you are correct. The reference to Table 4 and Figure 10 should be Table 5 and Figure 11. We have made the necessary corrections.

Comment10:

Table 5. The title needs to better describe the data in the table. I think the title could be “difference between automated and manual detection of grooming behaviors “. What does “number of matching results” mean. Remove” manual statistics” from the title. Use “manual detections” instead. This section needs to be described better.

Response10:

Thank you for your suggestion! We have revised the title of Table 5 to better describe the data in the table, as suggested. The title now reads: " Differences between our method and manual observation." The header has also been modified for clearer expression.

We would like to extend our heartfelt gratitude to you, the reviewers, for your invaluable feedback and insightful comments on our manuscript. Your expertise and constructive suggestions have played a pivotal role in improving the quality and clarity of our research. We sincerely appreciate the time, effort, and dedication you have invested in reviewing our work.

Best regards,

Wei Zhan

Round 2

Reviewer 1 Report (Previous Reviewer 2)

I made the following remarks to the basic version of the article:

1. First of all, the content of the article does not match its title. There is simply no "Statistical Mechanism" in the article. Moreover, all ten formulas given in the article are well known (references should be added). As far as I understand, "Real-time" is provided by high-speed shooting. The model used for recognition and detection is in no way adapted for "Real-time".

2. There is simply no scientific novelty in the work. The work offers nothing new in either Statistical or Real-time or Recognition and Detection. Moreover, I am sure that the system will not distinguish "Bactrocera minax" from an ant with wings. Thus, it is not an "Article". I think it's a Report.

3. Many questions arise about the rationale for the tools chosen by the authors. For example, if the authors say "Real-time", then I will answer "YOLO". The YOLO model directly predicts bounding boxes and class probabilities with one network in one estimate. The simplicity of the YOLO model allows for real-time predictions. Initially, the model takes an image as input. He divides it into an SxS grid. Each cell in this grid predicts B bounding boxes with a confidence score. This confidence is simply the probability of finding the object times the IoU between the predicted and ground truth fields. The CNN used is inspired by the GoogLeNet model, which introduces initial modules. The network has 24 convolutional layers followed by 2 fully connected layers. Reduction layers with 1x1⁴ filters followed by 3x3 convolutional layers replace the original seed modules. The Fast YOLO model is a lighter version with only 9 convolutional layers and fewer filters. Most of the convolutional layers are pre-trained using the Classified ImageNet dataset. Four convolutional layers are added to the previous network, followed by two fully connected layers, and it is fully retrained with the 2007 and 2012 PASCAL VOC datasets. The last layer outputs a tensor S*S*(C+B*5) corresponding to the predictions for each grid cell. C is the number of estimated probabilities for each class. B is a fixed number of anchor boxes per cell, each of these anchor boxes is associated with 4 coordinates (box center coordinates, width and height) and a confidence value.

4. The authors say “Ji [36] first proposed using 3D convolutional neural networks for human behavior 68 recognition. 3D convolutional neural networks have the advantage of simple network 69 structure, it can be executed end-to-end [37]. The subsequently emerged C3D [38] model 70 has also achieved better results in human behavior recognition.” I agree in principle with what was written, but why did the authors decide that a model that worked well in classifying human behavior would be effective in classifying insect behavior? Dear authors, the power of such neural networks lies in pre-training on colossal specialized datasets. In these datasets, insects were absent as a class.

The authors responded to my comments. I liked the resourcefulness of the authors, which manifested itself in their answers. I still believe that the scientific novelty of the article material is limited. At the same time, I cannot fail to note the obvious practical value of the proposed model. In general, I recommend the article for publication. I wish the authors creative success.

-

Author Response

Dear Reviewer1:

Thank you for highlighting the shortcomings in our paper. We have revised the abstract and conclusion sections to highlight our contributions and innovations. We hope that these changes in the revised manuscript will provide readers with a clearer understanding of our work upon their initial reading.

In the revised manuscript, we have emphasized the modifications made to the C3D network and provided statements regarding the experimental results. In doing so, readers will gain a better understanding of our work.

The latest version of abstract and conclusion and conclusion are as follow:

Abstract: Pest management has long been a critical aspect of crop protection. Insect behavior is of great research value as an important indicator for assessing insect characteristics. Currently, insect behavior research is increasingly based on the quantification of behavior The traditional manual observation and analysis methods can no longer meet the requirements of data volume and observation time. In this paper, we propose a method based on region localization combined with improved 3D convolutional neural network for six grooming behaviors of Bactrocera minax: Head grooming, Foreleg grooming, Fore-mid leg grooming, Mid-hind leg grooming, Hind leg grooming, and Wing grooming. The overall recognition accuracy reached 93.46%. We compared the results obtained from the detection model with the manual observations, and the average difference was about 12%. This shows that the model has reached a level close to the manual observation. Additionally, the recognition time using this method is only one-third of that required for manual observation, making it suitable for real-time detection needs. Experimental data demonstrates that this method effectively eliminates the interference caused by the walking behavior of Bactrocera minax, enabling efficient and automated detection of grooming behavior. Consequently, it offers a convenient means for studying pest characteristics in the field of crop protection.

Conclusion

This paper presents a statistical mechanism that combines insect body region localization and an improved C3D network for the recognition of Bactrocera minax grooming behavior. The proposed method involves precise cropping of Bactrocera minax body parts using the YOLO algorithm. The cropped consecutive frames are then stacked and inputted into the improved 3D convolutional neural network. Through scale fusion, the network combines deep and shallow features to perform classification and interval record for grooming behavior. Experimental results demonstrate the method's efficiency, accuracy, and real-time capability in detecting Bactrocera minax grooming behavior in videos, with errors falling within an acceptable range. This approach provides a novel means to analyze insect behavior in the context of precision agriculture.

Once again, we sincerely appreciate your thorough review and guidance. We hope the revisions we have made can improve the quality of the paper. Your constructive suggestions have played a pivotal role in improving the quality and clarity of our research. We sincerely appreciate the time, effort, and dedication you have invested in reviewing our work.

Best regards,

Wei Zhan

This manuscript is a resubmission of an earlier submission. The following is a list of the peer review reports and author responses from that submission.

Round 1

Reviewer 1 Report

This paper proposes a new algorithm to detect the grooming behavior of Bactrocera minax (Diptera: Trypetidae), which can effectively avoid the interference of Bactrocera minax walking behavior on its grooming behavior and reach several times the speed of human observation of this behavior. This approach contributes well to improve the research level of entomology, which aligns with the theme of the progress of computer vision in precision agriculture. Some suggestions are given as following..

1The Column width of Table 4 should be reset.

2Row 388 and 389 should be consolidated into one row.

3The indentation of letters in 144 rows (x1,x2,x3,x4) is wrong.

4Equations 1-6 are bolded, but the boldness is not necessary.

5Fig 10. The confusion matrix should be normalized.

6The authors should point out the contribution of this paper to smart agriculture and precision agriculture in conclusion.

7Some references are too old, and the authors should use new references whenever possible.

Not bad.

Reviewer 2 Report

The article is devoted to the applied application of 3D Convolution Neural Network. The topic of the article is relevant. The structure of the article is classical for MDPI (Introduction, Models and Methods, Experiments, Discussion, Conclusions). The level of English is acceptable. The article is easy to read. The quality of the figures is acceptable. The article cites 44 sources, some of which are not relevant. The References section is sloppy.

The following comments and recommendations can be formulated on the material of the article:

1. Data is one of the most important components of machine learning. The more of them, the better we can train our model, the more it will have a generalizing ability. For the field of 3D ML, today there is no single form of data representation that would be both compact, computationally efficient and easily extracted from real data. For example, there are polygonal models which are characterized by such shortcomings: - The need to create a special mathematical apparatus for extracting features from polygonal models, for example, convolutions on graphs; - Sensitivity of the format to data outliers. There are voxels. Their shortcomings: - “Roughness” of approximation of the shape of real objects at low resolution; - The amount of memory occupied by voxel models grows cubically depending on the number of intervals into which each axis is divided. There are point clouds. Their shortcomings: - Data disorder leads to the problem of choosing the error function and the need to develop a special mathematical apparatus; - There is no information about the connections between points, which does not allow to correctly restore the geometry and topology of the object under study and requires a complex post-processing procedure to translate the result into other forms of spatial data representation. Another well-known functional approach to the presentation of 3D data. Its disadvantages include: - A small number of datasets (training samples) using this format; - Any other formats are extremely problematic to bring to a functional; - Difficult to work with textures and shaders. So what model of 3D data representation did the authors choose? What are his shortcomings? How do the authors propose to mitigate these restrictions?

2. 3D ML is a separate area. It uses its own qualitative metrics. The Intersection over Union (IoU, also known as the Jaccard index) metric is a number from 0 to 1 that shows how much the internal “volume” of two objects (reference (ground true) and current) matches. In order to calculate $IoU$, it is necessary to be able to calculate the internal volume of the objects under consideration. In cases with polygonal models, the Monte Carlo method is most often used to estimate the volume. The Chamfer loss/distance metric is used to work with both polygonal models and point clouds. It shows how close the vertices of one polygonal model (point cloud) are to the vertices of another polygonal model (point cloud), and therefore should be minimized. Usually, a polygonal model obtained as a result of the algorithm is compared with a similar model from the dataset. The Normal loss / distance metric is similar to the previous one, where we consider two point sets P and Q, but in addition to the information about the vertices, we also use information about the normals. The disadvantage of chamfer and normal loss is the sensitivity to outliers. In addition to sensitivity to outliers, polygon overlap effects are often observed in practice. To avoid this effect, special shape regularizers are used along with chamfer and normal loss. In order for the final models to have smoother surfaces without emissions and noise, smoothing regularizers are often used. The simplest smoothing regularizer is Smooth loss. Another example of a smoothing regularizer is the Laplacian loss. Another commonly used metric, mainly for point clouds, is the Earth mover's distance (EMD), also known more generally as the Wasserstein metric. In addition to the metrics listed above, there are others. So, for example, for voxel representation of models, voxels are often “stretched” into ordinary numeric vectors and use metrics and loss functions for vectors, such as a cosine measure. After the model is trained and shows an acceptable result in terms of quality metrics, it needs to be built into the general data processing pipeline in a particular project. At this stage, other indicators of the quality of the model used appear, such as: the size of the memory that the model occupies; the amount of memory required for data processing by the model; speed of data processing by the model; the time required for re/additional training of the model. All these characteristics are referred to as external quality metrics, while the metrics discussed above are referred to as internal quality metrics of the problem being solved. It is also important that the resulting architecture and the way it is trained are reproducible. Usually, in articles describing a new architecture, only the learning algorithm, its parameters, and on which dataset the training was performed are indicated. All this should be taken into account by the authors.

3. Convolution is a wonderful tool that allows you to “accumulate” information about nearby (local) parts in the data. In the case of images, we naturally assume that adjacent pixels are likely to be semantically related (part of the same object). Since the same convolution is applied to all pixels in the image, the operator based on it becomes independent of the image size and takes up less memory. Based on this consideration, it becomes easy to understand why convolutional networks have become so popular for image processing. This raises the question: will convolution be applied to Euclidean or non-Euclidean data? Classify your data according to the criterion, dear authors. Let me remind you the properties of non-Euclidean data:

- Shift-invariance: if the object of interest (for example, a car) in the image is not in the upper left part, but in the lower right, then the convolutional filter will still be able to determine the location of the car. A little more formally, this data property allows the same operator to be applied to all parts of the data.

- Locality of data (locality): a group of nearby pieces of data is likely to be responsible for a certain object (in the case of a car, this could be a group of pixels responsible for a wheel or a car number). Formally, this allows us to apply a relatively small convolution size in order to localize the object of interest in the images.

- Data hierarchy (compositionality or hierarchy): large-scale and small-scale structures within one data object, most likely, are semantically related by the relation “is an integral part” (for a car, a group of pixels that defines a car consists of smaller groups that define car components: headlights, hood , wheels, etc.).

-

Reviewer 3 Report

Throughout: species names must be in italics

Inconsistent spacing and location of citation number

L15 states Drosophila citrus

L16 method behavior: should this be “method of analysing behavior”?

L50 “it” should be replaced with the species name

L55 mentions “combs”. I don’t know what they are referring to.

L76 need to explain further what is meant by “with a smaller size and faster behavior frequency”

L81 refers to carapace region of the insect. There is no structure in insects that is referred to as the carapace. Also the authors refer to Drosophila melanogaster which is very different to and much smaller than Bactrocera. I 

L94 refers to Citrus macrophytes: I don’t know of any species with that name

L97–98 needs rephrasing

L170-179 name the legs properly 

Behaviours are not defined they are just stated

Need to cite literature on Drosophila grooming such as work by Seeds et al. "A suppression hierarchy among competing motor programs drives sequential grooming in Drosophila." Elife 3 (2014): e02951.

Carapace maybe means thorax?

L239 depth is deepened

L428 – 429 “such as Drosophila melanogaster, Drosophila tangerine, Drosophila zeylanica, and Drosophila melanogaster” melanogaster is repeated. To my knowledge there is no species called Drosophila zeylanica or Drosophila tangerine

It is very difficult to know what the authors did, and what species they worked on. The expression in the paper is very unusual. It gives me the impression that a translation app combined with AI-based writing has been used. For example, the same leg structures are given different names in different parts of the text. They refer to "combs" which could be derived from a poor translation of "grooming".